# Personalized Therapeutic Strategies in the Management of Osteoporosis in Patients with Autoantibody-Positive Rheumatoid Arthritis

**DOI:** 10.3390/jcm11092341

**Published:** 2022-04-22

**Authors:** Bernardo D’Onofrio, Michele di Lernia, Ludovico De Stefano, Serena Bugatti, Carlomaurizio Montecucco, Laura Bogliolo

**Affiliations:** 1Division of Rheumatology, IRCCS Policlinico San Matteo Foundation, 27100 Pavia, Italy; michele.dilernia01@universitadipavia.it (M.d.L.); ludovico.destefano01@universitadipavia.it (L.D.S.); serena.bugatti@unipv.it (S.B.); carlomaurizio.montecucco@unipv.it (C.M.); l.bogliolo@smatteo.pv.it (L.B.); 2Department of Internal Medicine and Therapeutics, University of Pavia, 27100 Pavia, Italy

**Keywords:** osteoporosis, anti-citrullinated antibodies, rheumatoid arthritis, denosumab

## Abstract

Bone mineral density (BMD) reduction and fragility fractures still represent a major source of morbidity in rheumatoid arthritis (RA) patients, despite adequate control of the disease. An increasing number of clinical and experimental evidence supports the role of autoantibodies, especially anti-citrullinated protein antibodies (ACPAs), in causing localized and generalised bone loss in ways that are both dependent on and independent of inflammation and disease activity. The human receptor activator of nuclear factor kappa B and its ligand—the so-called RANK-RANKL pathway—is known to play a key role in promoting osteoclasts’ activation and bone depletion, and RANKL levels were shown to be higher in ACPA-positive early untreated RA patients. Thus, ACPA-positivity can be considered a specific risk factor for systemic and periarticular bone loss. Through the inhibition of the RANK-RANKL system, denosumab is the only antiresorptive drug currently available that exhibits both a systemic anti-osteoporotic activity and a disease-modifying effect when combined with conventional synthetic or biologic disease-modifying anti-rheumatic drugs (DMARDs). Thus, the combination of DMARD and anti-RANKL therapy could be beneficial in the prevention of fragility fractures and structural damage in the subset of RA patients at risk of radiographic progression, as in the presence of ACPAs.

## 1. Introduction

Rheumatoid arthritis (RA) is a chronic, inflammatory, progressive disease which, if left untreated, can lead to joint destruction and disability [1]. Localized bone loss and the development of joint erosions is a hallmark of RA, especially in the presence of rheumatoid factor (RF) and/or autoantibodies against post-translationally modified proteins, such as anti-citrullinated protein antibodies (ACPAs) [2,3]. In addition, it is well known that RA patients are at increased risk for systemic bone loss, as expressed by a reduction in bone mineral density (BMD), and, therefore, for suffering from osteoporosis (OP)-related fractures two times more frequently than age- and sex-matched controls [4,5,6,7,8,9,10]. This extremely relevant comorbidity is due not only to the use of glucocorticoids, the loss of mobility and the chronic inflammatory status, but also to the presence of autoantibodies themselves, as has emerged from several experimental lines and clinical evidence [11,12]. Many authors have highlighted the association between the presence of ACPAs (and, maybe, the RF) and lower BMD values at baseline in patients with early RA [13,14]. Rather, it is less clear if the abrogation of inflammation by the prompt institution of therapy with disease-modifying anti-rheumatic drugs (DMARDs) in these patients is enough to prevent the risk of generalized bone loss, or if the presence of ACPAs alone is sufficient to induce a reduction in BMD regardless of the coexistence of other risk factors, especially for the status of remission or low disease activity (LDA) [15,16].

The prognosis of RA has radically changed since the introduction of modern treat-to-target (T2T) and tight-control (TT) strategies; furthermore, the prompt institution of immunomodulating therapies has led to an improvement in the quality of life and a significant reduction in comorbidities as well as the risk of extra-articular manifestations [17,18,19]. OP-related fragility fractures, however, remain one of the most severe complications in RA patients, contributing to pain, a reduction in life expectancy and, ultimately, mortality [20,21].

The treatment of RA-related OP is mainly based on two classes of drugs: anabolic and antiresorptive drugs. The former includes teriparatide, a recombinant form of human parathyroid hormone (that is, the 1-34 aminoacidic fragment and the N-terminal biologically active portion) [22], which stimulates new bone formation. Bisphosphonates and denosumab reduce bone resorption through osteoclasts inhibition in a variable targeted way [23,24]. Denosumab is a fully human monoclonal IgG2 antibody that specifically binds to human receptor activator of nuclear factor kappa B ligand (RANKL), which belongs to the tumor necrosis factor superfamily, capable of inducing a reduction in the survival and activity of osteoclasts, and, therefore, a decrease in bone resorption. In light of the different mechanisms through which RA patients may experience a more accelerated bone loss and considering how the presence of ACPAs can represent an additional risk factor, probably at least in part due to their indirect action in the RANK-RANKL system, together with the well-known role of these autoantibodies in inducing erosions, some Authors have speculated on the possibility of a preferential choice of denosumab in the subgroup of seropositive patients [25,26,27,28].

The purpose of this review is to evaluate the opportunity to provide a tailored therapeutic choice for the management of OP in ACPA-positive RA patients.

## 2. Epidemiology of OP and Fragility Fractures in RA: Still an Unmet Need

Systemic bone loss is a common feature in RA patients and, as noted above, decreased mean BMD values in the hip and/or lumbar spine measured by dual-energy X-ray absorptiometry (DEXA) are more frequent in these patients compared with sex- and age-matched healthy controls [4,5,6,7]. At the same time, clinical studies that analysed BMD loss in the hip and spine in early RA patients showed an association between reduced BMD values and disease severity, in terms of laboratory findings (i.e., acute phase reactants) and progressive radiological damage and disability [29]. Furthermore, the measurement of BMD in the hands by digital X-ray radiogrammetry (DXR) shows an even more prominent depletion in bone at this level [30]. These findings suggest the presence of a common pathophysiological pathway in generalized as well as localized bone loss and erosions, especially triggering the RANK-RANKL system. Established RA patient show a prevalence of BMD loss doubled compared with healthy controls, as emerged from observational and case-control studies. Nevertheless, this phenomenon is known to start early in course of the disease, and after a two-year duration from the onset, patients experience significant bone loss in the hip and spine in all locations; notably, many authors observed a remarkable, more severe reduction in BMD ACPA-positive early RA patients, suggesting a direct effect of autoantibodies in bone re-modelling [31].

The direct consequence of these observations is that RA patients experience a higher rate of fragility fractures than the general population, and most epidemiological studies provide an overall fracture risk that is increased by from 1.5- to 2-fold among patients with RA compared to healthy controls (vertebral, hip, non-vertebral and non-hip fractures). Older age, white race, high as well as low glucocorticoid daily dose, prior fractures, longer disease duration and high disease activity are known to be risk factors for vertebral OP fractures, other than traditional ones such as smoking, alcohol abuse, sarcopenia, and a sedentary lifestyle [32,33,34]. Many studies have investigated the frequency of vertebral fractures in the RA population, with controversial results according to the different patients’ cohorts analysed, and the range varies from 8 to 49% of patients. Moreover, Guañabens and colleagues have recently highlighted that vertebral OP fractures in RA post-menopausal women increase over the years despite therapeutic advances, suggesting that the achievement of remission or LDA thanks to T2T and TC strategies is useful, although it is not enough to reduce fracture risk [35], and that risk seems to be high even for other, non-vertebral, fractures [36].

Similarly, the incidence rate of non-vertebral osteoporotic fractures is higher in RA patients than matched controls, with about 10 cases/1000 person per year [9]. Among them, hip fractures are the most severe in terms of morbidity and mortality, and they are 1.5–3 times more frequent in RA patients compared to healthy controls, as has emerged from different studies [37,38]. Similarly to vertebral fractures, major risk factors for hip fragility fractures are female sex and post-menopausal age, long disease duration, high disease activity and high functional impairment, as expressed by high Health Assessment Questionnaire (HAQ) values [39].

Data on non-vertebral and non-hip OP fractures (i.e., OP minor fractures) are less known and still quite controversial. Ørstavik et al. [5] did not find statistically significantly higher rates of self-reported humerus, wrist and ankle fractures among RA patients compared to healthy controls, and lower BMD was the unique independent risk factor associated with them. A more recent prospective study by Ochi and colleagues [40] observed a correlation between proximal humerus fractures with older age, glucocorticoid use and history of prior fractures. The same authors found an association between distal radius and female gender, older age, daily prednisolone dose and physician global visual analogue scale [41]. Vis and colleagues [42] performed a 5-year follow-up study of postmenopausal women with established RA and found a high incidence of vertebral and non-vertebral fractures (upper arm, wrist, hip, upper leg, ankle, ribs and pubic bone); furthermore, the presence of baseline non-vertebral fractures was an independent risk factor for new vertebral fractures.

In summary, the major, specific, risk factors for OP fractures in RA patients are glucocorticoid use, long disease duration, disease activity and disability. Of note, none of these studies highlighted an association between autoantibody status and fragility fractures. This could be explained by the observation that osteoporotic fracture is a multifactorial process, whereby the reduction in BMD is actually just one of the many contributing factors, and so it can be hard to find a direct association between autoantibodies and the fratturative event; however, further studies are required to better investigate a possible correlation [43].

## 3. Autoimmune-Induced Bone Loss in RA: The Role of ACPAs

In RA, inflammatory cytokines mediate the activation of osteoclasts and the concomitant inhibition of osteoblasts, resulting in a net loss of bone density both locally, in the form of erosions and periarticular osteopenia, and systemically, in the form of osteoporosis [44] Moreover, an increasing number of studies are defining the role of autoimmunity in determining bone resorption in RA in ways that are, at least partly, independent of inflammation [45].

Seropositivity for RF and/or ACPAs not only has diagnostic significance in RA, but also prognostic significance, because these antibodies tend to associate to a more severe disease progression. ACPAs are a class of antibodies directed against peptides in which arginine residues have been post-translationally modified to citrulline. Despite associating to increased structural progression of the disease, ACPA-positivity appears to associate to lower disease activity [3,46,47]. This so-called “ACPA paradox” suggests a pathogenic role of ACPAs that is independent of disease activity, and cortical and trabecular bone alterations have been demonstrated in ACPA-positive healthy individuals in the absence of clinical signs of arthritis [31,48]. The pathogenic effects of ACPAs on bone tissue do not appear to be limited to local bone loss; there are also effects on systemic bone loss, and anti-citrullinated vimentin antibodies are associated with increased serum RANKL levels in early untreated RA patients and seem to be capable to induce osteopenia when injected into mice [48,49].

It was indeed shown that the Fab fragment of ACPAs can directly bind to citrullinated vimentin on the surface of osteoclasts and monocyte-macrophage precursors, triggering osteoclast differentiation and activation by means of an interleukin (IL)-8-mediated autocrine loop [50,51]. Moreover, since ACPAs belong to the IgG subclass (mostly IgG1), their Fc tail can interact with the activating or inhibitory Fcγ receptors (FcγR) of various cell types, and osteoclast activation also occurs when immunocomplexes containing ACPA and citrullinated proteins bind to FcγR on the surface of mature and immature osteoclasts [52,53,54,55]. This interaction appears to be influenced by the glycosylation pattern of the Fc fragments: low sialylation is related to increased osteoclast formation, while increased sialylation abrogates the pro-osteoclastogenic activity of ACPAs [56]. Also, ACPAs may cause osteoclastogenesis indirectly, by triggering TNF release from macrophages and monocytes by binding to GRP78 on their surface and via FcγR-mediated activation [57,58,59]. Potentiation of the effects of ACPAs on osteoclastogenesis can be observed in the presence of RF, likely as the result of immune complexes formation with ACPAs, enhancing ACPAs ability to stimulate osteoclast activation and cytokine production in macrophages [13,60,61,62,63].

Anti-carbamylated protein antibodies (anti-CarP), although uncommon in clinical practice, also appear to be involved in mediating bone resorption in RA and have been correlated to periarticular hand and foot erosions. Recent in vitro studies have shown that anti-carbamylated-histone-IgG complexes and anti-carLL37-IgG immune complexes potentiate osteoclasts formation and activity [64,65].

Furthermore, while stimulation with osteoclastogenic cytokines causes the expression of RANKL in normal B cells, the spontaneous expression of RANKL was observed in memory and effector B cells from patients with RA, which appears to be triggered as a result of B cell receptor activation, rather than due to inflammatory stimuli [66,67].

An increasing number of studies support the direct role of autoimmunity on generalised bone loss in RA. In 2016, we published a study based on a cohort of 155 treatment-naïve patients with early RA, which first demonstrated an independent correlation between ACPA-positivity and reduced densitometric values in patients with early RA [13], thus providing additional support to the osteoclastogenic effects of ACPAs that were previously observed in vitro and in animal models. ACPA-positive patients showed significantly lower Z-scores in the lumbar spine compared to ACPA-negative patients, and the concomitant presence of RF seemed to potentiate the effects of ACPA in a dose-dependent manner. A significant difference in hip Z-score was also observed, but only in association with high titres of ACPA and RF. These findings have been supported by subsequent studies on other early RA cohorts. A study of Bruno et al. [14] has demonstrated an increased prevalence of osteopenia and osteoporosis in ACPA-positive vs. ACPA-negative patients in a cohort of 71 early RA patients that underwent femoral or lumbar spine DEXA.

Similar findings were recently published by Amkreutz et al. [15] on two separate early RA cohorts from the Netherlands and Sweden. ACPA positivity was associated with significantly lower baseline absolute BMD and Z-scores values both at the lumbar spine ad at the left hip in the Dutch cohort. Although the differences in BMD and Z-scores between ACPA-positive and ACPA-negative patients failed to reach statistical significance in the Swedish cohort, a higher prevalence of osteopenia was found in ACPA-positive patients at baseline. No independent association was found between RF and anti-CarP status and lower BMD levels at baseline. While it is quite clear that autoantibodies associate with bone loss at RA diagnosis, their role in determining bone loss in the long term is less clear. In the aforementioned study by Amkreutz and colleagues [15], ACPA positivity did not associate to a greater decrease in absolute BMD values, Z-scores or greater incidence of osteopenia over a follow-up period of 5 years for the Dutch cohort and 10 years for the Swedish cohort. Even when considering patients with higher titres of ACPA and patients showing higher disease activity during the first two years of the study, as expressed by disease activity score >1.8, no association was found between ACPA positivity and greater BMD loss over time. However, we have recently published a longitudinal analysis based on a cohort of 100 early RA patients and demonstrated a significant reduction in BMD values at both the lumbar spine and the femoral neck after two years of T2T [11]. After adjusting for other covariates such as age, disease activity, glucocorticoid and bisphosphonate usage, ACPAs still maintained an independent association with increased bone loss at the lumbar spine, but not at the femoral neck. ACPA-positive patients also showed smaller increases in BMD after treatment with bisphosphonates compared to ACPA-negative ones. No clear association was demonstrated between bone loss and higher ACPA levels or RF positivity. Moreover, another study by Tomizawa et al. [68] demonstrated that ACPAs are a significant predictor of annual BMD change at the proximal femur in a cohort of 214 established RA patients treated to target after a 2-year observation period.

As a retrospective observation, recent studies have shown that the use of therapies inducing the depletion of B lymphocytes and, therefore, a reduction in the levels of autoantibodies also had a favorable impact on BMD preservation; however, whether this effect is also due to the inhibition of the direct pro-osteoclastogenic activity of B cells is still to be clarified [69]. Moreover, other studies compared BMD loss reduction derived from the inhibition of T cell co-stimulation, in comparison with conventional synthetic DMARDs and TNFα-inhibitors, and observed a bone-sparing effect of abatacept in RA patients; none of the patients in the cohorts were treated with rituximab [70,71].

## 4. Denosumab as a Disease-Modifying Anti-Rheumatic Drug

Targeting the RANK-RANKL system in an indirect (and, maybe, direct) fashion by ACPAs results in osteoclasts prolonged survival and aberrant activation, with systemic bone loss, BMD reduction and the development of OP. At the same time, it is well known that seropositivity strongly influences the possibility of developing periarticular osteopenia and joint erosions in RA patients, which is at least partly accounted for ACPAs’ role in stimulating the RANK-RANKL system [3]. Thus, the use of denosumab in seropositive-RA patients could be doubly beneficial, both in managing OP and reducing local bone loss and radiographic progression.

Several studies have investigated the possible activity of denosumab as a disease-modifying drug (Table 1). A multicenter phase II study performed by Cohen and colleagues in 2008 [72] aimed to evaluate the effect of denosumab at a dosage of 60 mg versus 180 mg versus placebo in terms of erosive progression, as assessed by magnetic resonance (MRI) and X-ray imaging of the hands in a cohort of RA patients treated with methotrexate. This revealed a reduced rate of structural damage in denosumab 180-mg group at 6 months and in 60 plus 180 groups at 12 months. Notably, the concordance between MRI and radiographic trends (especially in the 180-mg-treated subgroup) highlighted the true anti-erosive effects of denosumab. These results were confirmed in subsequent studies, which also showed a decrease in bone loss in the hands as measured by DXR; no beneficial effect was noted in terms of reductions in joint space narrowing [73,74,75]. In addition, the DRIVE trial, published in 2016 by Takeuchi and colleagues [76], which stratified RA patients for RF positivity and glucocorticoid use, pointed to the possible utility of denosumab in reducing joint erosions, especially in those subjects with risk factors for radiographic progression. A recent study also evaluated its beneficial effect in preventing joint erosions in RA patients with regards to the presence of ACPAs, confirming a good efficacy profile [77]. Of note, and as expected, none of the patients in the DRIVE trial (all of them were treated with methotrexate according to the T2T strategy) who were randomly assigned to subcutaneous injections of placebo or denosumab 60 mg every 6 months, every 3 months, or every 2 months showed any differences in terms of the American College of Rheumatology response or disease activity score 28, suggesting a substantial lack of efficacy of denosumab in influencing almost all the core domains of disease activity. Given its mechanism of action, which does not have an impact on the inflammatory and cytokine pathways, denosumab was not expected to have a substantially positive action in disease activity reduction since is not able to extinguish the phlogistic process. Thus, denosumab is not intended to be used as a substitute for a cs- or bDMARD in the treatment of RA, but as an adjunct to the traditional therapy.

There are few data on the possible disease-modifying action of other antiosteoporosis drugs. Concerning bisphosphonates, zoledronate is the most efficacious drug of this group, capable of inducing apoptosis of the osteoclasts [84] and, at least in an experimental setting, collagen-induced arthritis in mice, reducing structural damage progression when associated with methotrexate [85]. However, prospective randomized trials failed to confirm a disease-modifying action of zoledronate in reducing bone destruction even if evaluated in monotherapy in erosive psoriatic arthritis patients [86] and in tophaceous gout [87]. Yue and colleagues [79] investigated the comparative efficacy of denosumab 60 mg in one injection and oral alendronate 70 mg per week in reducing the size of pre-existing bone erosions in RA; they observed a bone-healing effect of denosumab after 6 months of treatment, which was not present in the alendronate-treated patients.

As expected, teriparatide failed to show significant effects on bone erosion of the hands or wrists in RA patients treated with TNF inhibitors and not taking other osteoporosis treatment [88]; these results were subsequently confirmed by Ebina et al. [81], who showed a positive effect on the prevention of bone erosions only when oral bisphosphonates were switched to denosumab in patients with RA, and not if they were continued or switched to daily teriparatide. The same observations were also validated by other authors [89].

No data are available concerning a possible disease-modifying role of romosozumab, a humanized monoclonal antibody that inhibits sclerostin and thus activates the Wnt signalling pathway resulting in osteoblast proliferation; it has a primary effect in stimulating bone formation and a minor, secondary effect in decreasing bone resorption [90,91]. A single randomized, double-blind, placebo-controlled study displayed no beneficial effect of romosozumab on the disease activity of RA patients with concomitant severe osteoporosis [92]. However, given its mechanism of action, romosozumab is unlikely to influence structural progression.

A novel, small binding-RANKL peptide called OP3-4 was described to promote both osteoclast inhibition and new bone formation in a murine collagen-induced arthritis model, and this property seems to be unique to this RANKL-inhibitor. However, it is actually not known whether OP3-4 systemically promotes bone formation, in specific locations or only in the presence of focal erosions; thus, its possible role in managing OP has yet to be determined [93,94].

In summary, by analysing the data derived from clinical trials as well as retrospective studies and reviews, denosumab seems to be the only currently available antiosteoporotic drug that can be useful in reducing not only systemic but also periarticular bone loss, conditions that are both strongly manifest in ACPA-positive RA patients.

## 5. Efficacy and Safety of Denosumab in the Treatment of OP in RA Patients

The same large, randomized trials that analysed the effect of denosumab on radiographic progression in RA also agree on its efficacy in systemic bone loss prevention. Moreover, denosumab can be considered as an option for the prevention and treatment of glucocorticoids-induced OP (GIOP), which afflicts a large portion of RA patients, even when compared to bisphosphonate agents such as alendronate and risedronate [95,96,97]. This was also confirmed by the observation of the bone turnover marker suppression in RA patients treated with denosumab, for both OP treatment-naïve patients and after switching from bisphosphonates [98,99,100].

A very recent Japanese report showed a comparable efficacy of romosozumab and denosumab in increasing BMD in RA patients even under steroid treatment within a short time, with bone turnover markers suppressed only by the latter. Romosozumab is likely to be a good option for managing OP in RA, but no data are currently available on its potential disease-modifying effect [101].

Notably, almost all studies displayed a good safety profile for RANKL inhibition when compared to placebo groups [28,72,74,75,76]. These data are of great importance, since denosumab is associated with a few adverse events, which could potentially be even more common in RA patients. In particular, through a combination of steroids and denosumab, together with their chronic inflammatory status, RA patients could be more prone to experiencing osteonecrosis of the jaw, an extremely severe adverse effect derived from the use of antiresorptive drugs, which is described to be more frequent denosumab-treated patients compared to bisphosphonate-treated OP patients [102]. In addition, other potentially denosumab-related complications that need to be considered in RA patients include widespread musculoskeletal pain, hypocalcaemia (not so uncommon, due to renal impairment and steroids usage), skin reactions, rebound-vertebral fractures after denosumab withdrawal and skin and diverticular infections. Regarding the latter, since RANKL is expressed on the surface of B- and T-cell and in lymph nodes, denosumab was initially thought to cause a potential increase in infections, especially when used in combination with immunosuppressive agents [103]. The FREEDOM trial, which first evaluated the use of denosumab for the treatment of osteoporosis in post-menopausal women, did not find an increased rate of infections in the denosumab-group compared to placebo [104], and these observations were subsequently confirmed in a subsequent analysis as well as in immunocompromised patients [105,106,107]. Moreover, a very recent metanalysis, which analysed the rate of adverse events in post-menopausal women with osteopenia/porosis treated with denosumab versus placebo, did not find an excess of side effects in the denosumab-treated group, with the exception of throat pain, constipation and skin rashes [108]. Of note, further studies are needed to evaluate the safety profile of denosumab in the specific subgroup of RA patients.

## 6. Conclusions

Systemic bone loss and fragility fractures still represent a major source of morbidity in RA patients, despite adequate control of disease activity. Besides the traditional risk factors, increasing evidence suggests that the presence of ACPAs could be a disease-specific condition, capable of worsening BMD reduction in these patients by indirectly triggering the RANK-RANKL system, and thus activating osteoclasts. At the same time, it has been broadly demonstrated that ACPAs represent a major risk factor for structural damage in RA patients. Unlike other classes of antiresorptive agents, denosumab can directly inhibit the RANKL and prevent bone resorption. Furthermore, an increasing number of studies suggest a concomitant, disease-modifying effect of denosumab when used in combination with conventional synthetic or biologic DMARDs. Altogether, in the era of personalized medicine, these observations can help to identify a subgroup of RA patients that could benefit from the combination of anti-RANKL therapy with a disease-modifying drug in RA management in the presence of bone loss.

## Figures and Tables

**Table 1 jcm-11-02341-t001:** The impact of Denosumab on erosions in rheumatoid arthritis (RA) patients.

Authors Country Year	Type of the Study	Studied Population	Study Design	Results
Cohen S [72] USA 2008	Phase II	218 RA patients on methotrexate therapy	75 patients received subcutaneous placebo, 71 denosumab 60 mg, 72 denosumab 180 mg injections every 6 months for 12 months	Twice-yearly injections of denosumab inhibited structural damage in patients with RA for up to 12 months
Sharp JT [75] USA 2010	Phase II	227 RA patients on methotrexate therapy	Patients were randomly located in a 1:1:1 ratio to receive denosumab 60 mg, 180 mg or placebo, at baseline and after 6 months	Twice-yearly injections of denosumab significantly reduced cortical bone loss in RA patients for up to 12 months
Deodhar A [73] USA 2010	Phase II	56 RA patients on methotrexate therapy	Patients were randomized in a 1:1:1 ratio to receive subcutaneous placebo, denosumab 60 mg, or denosumab 180 mg at 0 and 6 months	Denosumab provided protection against erosion, and not only prevented bone loss but increased hand BMD as measured by DXA
Takeuchi T [76] Japan, USA, The Netherlands 2016	Phase II	350 RA patients on methotrexate therapy, stratified for glucocorticoid use and seropositivity for rheumatoid factor	Randomly assigned to subcutaneous injections of placebo or denosumab 60 mg every 6, 3 or 2 months	Denosumab significantly inhibited the progression of bone erosion at 12 months compared with the placebo
Hasegawa T [78] Japan 2017	Retrospective	80 RA patients	40 RA patients treated with biologic disease modifying anti-rheumatic drugs (bDMARDs) plus denosumab and 40 RA patients treated with bDMARDs without denosumab	Concurrent use of denosumab and bDMARDs was efficacious in inhibiting structural damage
Yue J [79] China 2017	Post-hoc analysis of randomized controlled trial	40 RA patients treated with conventional synthetic (cs) DMARDs or bDMARDs	Randomized in a 1:1 ratio to receive either subcutaneous denosumab (60 mg) once or oral alendronate (70 mg) weekly for 6 months	Denosumab can induce partial repair of erosions in patients with RA, while erosions continued to progress in patients treated with alendronate
Mochizuki T [80] Japan 2018	Prospective	70 RA patients treated with DMARDs	All patients were administered denosumab 60 mg subcutaneous injection at baseline and at 6 months	Denosumab increased the BMDs of the lumbar spine, total hip, femoral neck and hand, and suppressed joint destruction of Japanese patients with RA
Ebina K [81] Japan 2018	Retrospective	90 RA patients in treatment with bisphosphonate (BP)	30 patients continued with BP treatment, 30 were switched to teriparatide, 30 were switched to denosumab	After 12 months, the mean changes of the modified Sharp erosion score were significantly lower in the switch-to-denosumab group compared to the other two groups
Ishiguro N [82] Japan, USA, The Netherlands 2019	Phase II	340 RA patients treated with methotrexate, stratified for RF and anti-citrullinated peptide antibodies (ACPA) positivity, swollen joint count, C-reactive protein and erythrocyte sedimentation rate level, disease duration and glucocorticoid use	Randomized to receive placebo or denosumab 60 mg every 6 months, 3 months or 2 months	Patients with risk factor for erosive disease showed consistent results for the change in the modified Sharp erosion score at 12 months from baseline
Takeuchi T [28] Japan, USA, The Netherlands 2019	Phase III	654 RA patients in therapy with csDMARDs	Randomly assigned (1:1:1) to denosumab 60 mg every 3 months, every 6 months or placebo	Denosumab groups showed significantly less progression of joint destruction assessed by the modified total Sharp score at 12 months
Mori Y [77] Japan 2021	Retrospective	106 ACPA-positive RA patients	All were previous treated with oral BP; 56 were switched to denosumab, 50 continued to be treated with BP	At 12 and 24 months, denosumab-group patients showed significant differences in the change in erosion score and modified total Sharp score
Tanaka Y [83] Japan, USA, The Netherlands 2021	Phase III	654 RA patients in therapy with csDMARDs	Randomly assigned (1:1:1) to denosumab 60 mg every 3 months, every 6 months or placebo	Denosumab groups showed significantly less progression of joint destruction assessed by the modified total Sharp score at 36 months

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
