# Peer review of "Personalized Therapeutic Strategies in the Management of Osteoporosis in Patients with Autoantibody-Positive Rheumatoid Arthritis"

_jcm, 2022, doi:10.3390/jcm11092341_

Round 1

Reviewer 1 Report

The authors carry out an interesting review about the association between RA and osteoporotic fractures and the possible role of ACPA+ in the pathogenesis of OP in RA, with the “Purpose of this review is to evaluate the opportunity of a tailored therapeutic choice for the management of OP in ACPA-positive RA patients” .

Their analyses points that the major, specific, risk factors for OP fractures in RA patients are glucorticoid use, long disease duration, disease activity and disability, and maybe ACPAs and RF.

In their review, the rationale for treatment with denosumab in RA is based on a drug capable of reducing radiographic progression of RA and do not analyze the reduction of the risk of osteoporotic fracture or improvement in BMD in the RA patient.

Ideally, their analysis should include the treatment of RA and the concomitant use of specific treatments of OP and their ability to reduce the fracture risk and show that denosumab is the best option of treatment.

If they do not include this analysis in their review, they cannot support their hypothesis

I recommend this review:

Raterman HG, Bultink IE, Lems WF. Osteoporosis in patients with rheumatoid arthritis: an update in epidemiology, pathogenesis, and fracture prevention. Expert Opin Pharmacother. 2020 Oct;21(14):1725-1737. doi: 10.1080/14656566.2020.1787381. Epub 2020 Jul 1. PMID: 32605401.

Author Response

To the Editor

To the Reviewers

We are grateful for considering our manuscript of interest and for providing a number of relevant comments and suggestions that have helped us improving the clarity of data presentation and discussion. We have taken all the Reviewers’ comments into great consideration and have extensively modified our manuscript accordingly, we sincerely hope that our work, in its current modified form, will be judged of value both in terms of novelty and focus, and will be considered suitable for publication in Journal of Clinical Medicine.

Kind regards,

Bernardo D’Onofrio

MD

Division of Rheumatology, IRCCS Policlinico San Matteo Foundation University Hospital, Viale Golgi

Reviewer 1

Comments to the Author

The authors carry out an interesting review about the association between RA and osteoporotic fractures and the possible role of ACPA+ in the pathogenesis of OP in RA, with the “Purpose of this review is to evaluate the opportunity of a tailored therapeutic choice for the management of OP in ACPA-positive RA patients”.

Their analyses point that the major, specific, risk factors for OP fractures in RA patients are glucorticoid use, long disease duration, disease activity and disability, and maybe ACPAs and RF.

Reply: We are grateful to the Reviewer for the interest in our review and for providing important suggestions which enabled us to broaden and improve our manuscript. We have taken all the comments into great consideration and have modified our work accordingly.

Query 1: Ideally, their analysis should include the treatment of RA and the concomitant use of specific treatments of OP and their ability to reduce the fracture risk and show that denosumab is the best option of treatment.

If they do not include this analysis in their review, they cannot support their hypothesis

Reply to Query 1: We thank the Reviewer, and we agree that the role of denosumab in osteoporotic fractures prevention and improvement of BMD in RA patients needs to be included in our work. We have now created a new paragraph discussing the topic (paragraph 5, pages 13-14 of the revised manuscript, track changes). We hope the Reviewer will agree with our data presentation.

Query 2: I recommend this review:Raterman HG, Bultink IE, Lems WF. Osteoporosis in patients with rheumatoid arthritis: an update in epidemiology, pathogenesis, and fracture prevention. Expert Opin Pharmacother. 2020 Oct;21(14):1725-1737. doi: 10.1080/14656566.2020.1787381. Epub 2020 Jul 1. PMID: 32605401.

Reply to Query 2: We thank the Reviewer for suggesting this comprehensive manuscript, which has now be included as reference 93 of the revised manuscript.

Reviewer 2 Report

The authors have collected a large number of relevant papers for their review in a comprehensive manner. My suggestion, however, would be to make the references logically instead of listing them one by one. In particular for those clinical trials reporting about the medicine, understand their internal differences, the core issues to be addressed, and then present them. Anyway, it would be a good review after minor revision.

Author Response

To the Editor

To the Reviewers

We are grateful for considering our manuscript of interest and for providing a number of relevant comments and suggestions that have helped us improving the clarity of data presentation and discussion. We have taken all the Reviewers’ comments into great consideration and have extensively modified our manuscript accordingly, we sincerely hope that our work, in its current modified form, will be judged of value both in terms of novelty and focus, and will be considered suitable for publication in Journal of Clinical Medicine.

Kind regards,

Bernardo D’Onofrio

MD

Division of Rheumatology, IRCCS Policlinico San Matteo Foundation University Hospital, Viale Golgi

Reviewer 2

Comments to the Author

The authors have collected a large number of relevant papers for their review in a comprehensive manner. My suggestion, however, would be to make the references logically instead of listing them one by one. In particular for those clinical trials reporting about the medicine, understand their internal differences, the core issues to be addressed, and then present them. Anyway, it would be a good review after minor revision.

Reply: We are grateful to the Reviewer for the interest in our review and for providing important suggestions which enabled us to broaden and improve our manuscript. We agree that our paragraph about the possible role of denosumab as a disease-modifying anti-rheumatic drug could be presented in a more analytic and resumptive way in order to better clarify the internal differences of clinical trials currently available. Since there are only few data about the topic, we decided to present them in a listed way. Anyway, for a better understanding, and thank to your useful comment, we added a brief summary in the end of the paragraph and we added a new paragraph as asked by the Reviewer 1 presenting the data in a more narrative way.

Round 2

Reviewer 1 Report

The review has improved substantially, mainly in efficacy.  The table 1 of effectiveness in erosion prevention improves reading.  

In the point 2. Epidemiology of OP and fragility fractures in RA about risk factors of FF, lines 124 to 126 should comment on why ACPAs have not been shown to be  a risk factor for  fractures. The phrase in lines 127 to 129 is too much and you should consider removing it. This is commented on in the next point lines 205 to 212 and is repetitive.

In addition, in the analysis of denosumab as DMARD emphasize the null effect of the drug on disease activity (lines 241 – 250). The effect is notably in erosion prevention and increase in DMO, but not in activity.

The security aspects deserve a review lines 312 to 315. Theoretical information on the possible beneficial effects of denosumab is extensive (Point 3 and Point 4 related to erosions). I consider that for the benefit of our patients, the safety section must be balanced. Specifically, in the data sheet of denosumab, product of the Phase III trials and post marketing setting (https://www.ema.europa.eu/en/medicines/human/EPAR/prolia) and a recent review in 2021 (https://www.ema.europa.eu/en/medicines/human/EPAR/prolia#product-information-section), issues appear on some points that should be considered in patients with RA:

  1. Generalized musculoskeletal pain
  2. Hypocalcemia particularly in patients with steroids and renal failure, both common in RA patients.
  3. Subcutaneous cell tissue infections also with increased risk in RA.
  4. Skin rash (hypersensitivity and lichenoid lesions).
  5. Diverticulitis may be a problem in patients with RA and some other biologics (Anti-IL6
  6. Jaw osteonecrosis
  7. Atypical femur fractures
  8. Multiple vertebral fractures after suspension of denosumab, considering that the risk of vertebral fractures is still one of the unmet needs in RA patients.

Best regards.

Author Response

To the Editor

To the Reviewers

We are grateful for considering our manuscript of interest and for providing relevant comments and suggestions that have helped us improving the clarity and usefulness of data presentation and discussion. We have taken all the Reviewer’s comments into great consideration and have modified our manuscript accordingly. We sincerely hope that our work, in its current modified form, will be judged of value both in terms of novelty and focus, and will be considered suitable for publication in Journal of Clinical Medicine.

Kind regards,

Bernardo D’Onofrio

MD

Division of Rheumatology, IRCCS Policlinico San Matteo Foundation University Hospital, Viale Golgi

Reviewer 1

Comments to the Author

The review has improved substantially, mainly in efficacy. The table 1 of effectiveness in erosion prevention improves reading. 

Reply: We thank the Reviewer for appreciating the modified version of our manuscript. We also thank for the further important suggestions which helped us to improve our review.

Query 1: In the point 2. Epidemiology of OP and fragility fractures in RA about risk factors of FF, lines 124 to 126 should comment on why ACPAs have not been shown to be a risk factor for fractures. The phrase in lines 127 to 129 is too much and you should consider removing it. This is commented on in the next point lines 205 to 212 and is repetitive.

Reply to Query 1: We thank the Reviewer for this important observation; accordingly, we provided to add a possible explanation for the not-described role of autoantibodies in increasing the risk of fracture events. We also thank for the suggestion of removing that repetitive phrase and we removed it.

Query 2: In addition, in the analysis of denosumab as DMARD emphasize the null effect of the drug on disease activity (lines 241 – 250). The effect is notably in erosion prevention and increase in DMO, but not in activity.

Reply to Query 2: We agree that the null effect of denosumab in influencing disease activity needed to be emphasised, in order to not to create misunderstandings for the readers: the RANK-L inhibition does not improve disease activity, though is not intended to substitute methotrexate and other DMARDs.

Query 3: The security aspects deserve a review line 312 to 315. Theoretical information on the possible beneficial effects of denosumab is extensive (Point 3 and Point 4 related to erosions). I consider that for the benefit of our patients, the safety section must be balanced.

Reply to Query 3: We thank the Reviewer for this comment which can help us to improve our manuscript considerably. We absolutely agree that the safety profile of denosumab IN RA patients should deserve a more extensive discussion; so, we modified our review accordingly.
